# Dispersion of a Traffic Related Nanocluster Aerosol Near a Major Road

**Oskari Kangasniemi [1,*] , Heino Kuuluvainen [1] , Joni Heikkilä [1], Liisa Pirjola [2,3], Jarkko V. Niemi [4], Hilkka Timonen [5] , Sanna Saarikoski [5], Topi Rönkkö [1] and Miikka Dal Maso [1]**

[1] Aerosol Physics Laboratory, Physics Unit, Faculty of Engineering and Natural Sciences, Tampere University, P.O. Box 692, FI-33014 Tampere, Finland; heino.kuuluvainen@tuni.fi (H.K.); joni.heikkila@tuni.fi (J.H.); topi.ronkko@tuni.fi (T.R.); miikka.dalmaso@tuni.fi (M.D.M.)

[2] Department of Technology, Metropolia University of Applied Sciences, P.O. Box 4021, FI-00180 Helsinki, Finland; liisa.pirjola@metropolia.fi

[3] Department of Physics, University of Helsinki, P.O. Box 64, FI-00014 Helsinki, Finland

[4] Helsinki Region Environmental Services Authority, P.O. Box 100, FI-00066 HSY Helsinki, Finland; jarkko.niemi@hsy.fi

[5] Atmospheric Composition Research, Finnish Meteorological Institute, P.O. Box 503, FI-00101 Helsinki, Finland; hilkka.timonen@fmi.fi (H.T.); sanna.saarikoski@fmi.fi (S.S.)

\* Correspondence: oskari.kangasniemi@tuni.fi

**Abstract:** Traffic is a major source of ultrafine aerosol particles in urban environments. Recent studies show that a significant fraction of traffic-related particles are only few nanometers in diameter. Here, we study the dispersion of this nanocluster aerosol (NCA) in the size range 1.3–4 nm. We measured particle concentrations near a major highway in the Helsinki region of Finland, varying the distance from the highway. Additionally, modelling studies were performed to gain further information on how different transformation processes affect NCA dispersion. The roadside measurements showed that NCA concentrations fell more rapidly than the total particle concentrations, especially during the morning. However, a significant amount of NCA particles remained as the aerosol population evolved. Modelling studies showed that, while dilution is the main process acting on the total particle concentration, deposition also had a significant impact. Condensation and possibly enhanced deposition of NCA were the main plausible processes explaining why dispersion is faster for NCA than for total particle concentration, while the effect of coagulation on all size ranges was small. Based on our results, we conclude that NCA may play a significant role in urban environments, since, rather than being scavenged by larger particles, NCA particles remain in the particle population and grow by condensation.

**Keywords:** nanocluster aerosol; dispersion; aerosol modelling

## 1. Introduction

Urban pollution has a significant impact on environment and health. A major part of this pollution is ultrafine aerosol particles originating from traffic [1]. Aerosols have an effect on the radiation balance and climate change [2], visibility [3], cloud formation [4] and the hydrological cycle [5]. High concentrations of aerosol particles have a damaging effect on respiratory and cardiovascular systems [6,7]. Urban aerosols have also been linked to elevated levels of harmful magnetite nanoparticles in the human brain [8]. Overall, it has been estimated that, globally, air pollution, consisting mainly of particles with diameter under 2.5 μm, causes over 3 million premature deaths annually [9].

　　　　　　

Recent measurements have shown that traffic is a major source of previously undetected particles in the particle diameter range of 1.3–3.0 nm [10]. This nanocluster aerosol (NCA) was also measured in a street canyon by Hietikko et al. [11], who noticed a clear connection between NCA concentration and traffic volume. Kontkanen et al. [12] reported measurements of sub-3 nm particles in different types of environments and observed higher concentrations in areas with higher anthropogenic emissions. Based on these studies, it is likely that NCA forms a significant fraction of the total particle number in an urban environment. Ultrafine particles have been speculated to have properties that elevate their hazardousness to human health, and the high concentration of NCA adds to the human exposure to nanosized particles in urban areas [13]. Nanocluster-sized particles also represent the first steps of the formation process affecting ultrafine particle number concentration: they have the potential to act as condensation seeds for low-volatility vapours and therefore impact the formation rate of secondary aerosols in urban areas.

The relative strength of traffic as a NCA source is still under investigation. In their study, Rönkkö et al. [10] measured NCA concentrations next to a main road in a semiurban environment and in a street canyon in an urban environment, both in the Helsinki region of Finland. They also measured NCA emissions in a long-distance on-road study using a mobile laboratory driven from northern Spain to Finland through western Europe. They concluded that the number of NCA emitted by traffic may exceed the number of new particles formed by atmospheric nucleation in urban environments. On the other hand, Yao et al. [14] observed that new particle formation in Shanghai correlates with strong solar radiation and high sulfuric acid and ozone concentrations indicating new particle formation was photochemically induced. The high concentration of existing particles in a heavily polluted environment should suppress new particle formation, but it has been seen that, in China, new particles are formed despite the heavy pollution [15]. In urban areas, neither source cannot yet be ruled out; however, their relative contribution in various urban settings remains unknown.

Particle concentrations originating from traffic depend heavily on the distance from a road acting as a source, and the wind conditions. Upwind concentrations tend to be significantly lower than downwind and in fact differ little from the urban background concentrations. Downwind concentrations are significantly higher next to the road and gradually decrease to urban background concentrations when moving further away from the road [16–18]. Downwind, emitted pollutants undergo dispersion, affected by wind speed, which has a significant effect on particle concentrations in urban environments and near roads.

Roadside number distribution measurements show number distributions with 2–3 modes. These modes are nucleation mode with peaks somewhere between 10–20 nm, Aitken mode (40–50 nm) and accumulation mode (100–150 nm) [16,17,19]. Nucleation mode particles are formed during the the dilution of the exhaust gas while larger primary or soot particles are formed during the combustion in the engine.

Understanding the evolution of the concentration of ultrafine particles and NCA as they are transported from their traffic source is a key piece of information to understand their relative contribution to the urban particulate loading. It is also needed to successfully incorporate observations of the emissions of these particles into regional and global modelling efforts, both for air quality and climate. While several current models are able to resolve particle number concentrations also at sizes close to the formation size, they rely on existing emission inventories for predictions of aerosol concentrations. Few inventories for particle number have been produced to date. For traffic, they rely strongly on the currently performed vehicle engine emission measurements, such as EURO 6 emission standard focusing on non-volatile particles larger than 23 nm [20,21]. This severely underestimates the number concentration of sub-23 nm particles. From a modelling point of view, these particles—at least partly formed during the early stages after entering the atmosphere—should be considered primary or delayed primary particles [10]. Delayed primary particles are defined as particles that are formed from low-volatility precursor gases in hot exhaust during fast cooling and dilution. They are therefore different from secondary particles, which are formed through oxidation and photochemical processes

in the atmosphere [10], even though their formation process is a gas-to-particle transition that is traditionally considered a secondary process. This distinction is important from a modelling point of view, as these particles are formed at a temporal and spatial scale below that of most 3D models.

After emission, fresh particles undergo transformation processes during dispersion. These processes cause the number distribution of the aerosol particles to evolve at varying rates, sometimes very rapidly. For point and line sources such as roads, dilution is the key process and has a significant effect on the other possible processes affecting the number distribution; it sets the timescale that other processes should be compared against. Other transformation processes acting on the aerosol population are nucleation, condensation, evaporation, coagulation, deposition and chemical reactions [22]. The effect of these processes varies between different particle size ranges. The main transformation processes and their effect on number concentrations of total particle population and NCA population have been shown in Figure 1, with the effect on total mass concentration also included. The relative magnitudes of these processes are the main factor determining the fate and contribution of emitted NCA in the urban environment.

Here, we present data of NCA measurements at varying distances from a roadway during a day with wind mostly perpendicular to the road, with the aim of describing the dispersion behaviour of nanocluster-size aerosol. We measured the total particle concentrations as well as the NCA concentrations at different distances from a source, which in this case is a busy ring road. To elucidate the effect of various dynamical processes on the particle size distribution, especially the NCA concentration, we simulated the aerosol dynamics in the diluting air using an aerosol dynamics model. As a result, we present to our knowledge the first measured concentration profiles of nanocluster aerosols near a roadway, and a quantitative description of the main processes that affect their concentration.

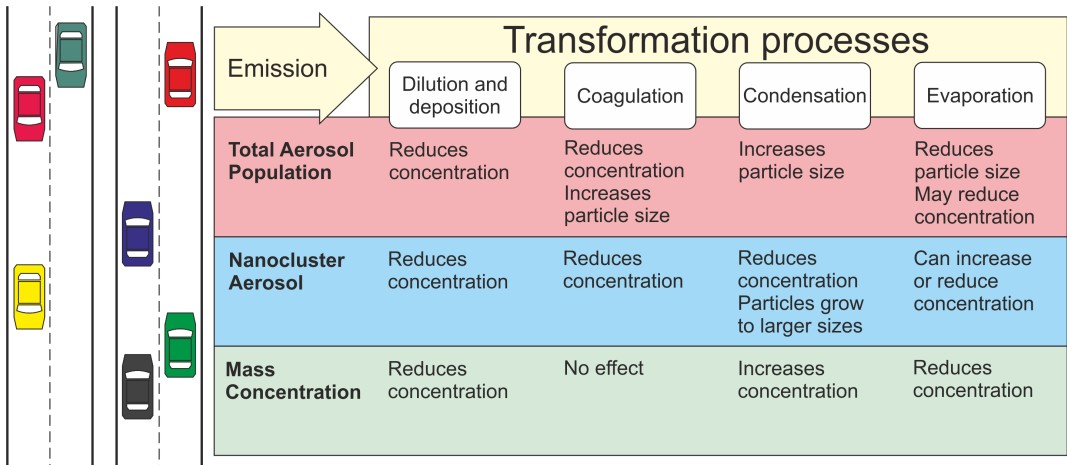

**Figure 1.** A schematic depicting the effect of dilution, deposition, coagulation, condensation and evaporation on the total aerosol particle number concentration, NCA number concentration and total mass concentration.

## 2. Methods

### 2.1. Experimental

The experimental part of this study consisted of mobile measurements of aerosol concentration using the Tampere University mobile observation laboratory equipped with instrumentation for quantification of submicron aerosol concentration in urban environments. In addition, another mobile laboratory from Metropolia ("Sniffer") was used as a stationary setup measuring wind conditions and aerosol particle concentrations. All of the measurements were performed on 17 April 2015 during both morning and afternoon. The measurement day was a Friday and the traffic was normal weekday traffic. The measurements were done next to Ring III, an important highway in the Helsinki region of

Finland. The average daily traffic in 2015 at the nearest counting point 900 m from the measurement site was 51,000 vehicles during weekdays, out of which 10.6% was heavy-duty vehicles [23]. The traffic volume did not vary significantly between the morning and the afternoon measurements with an hourly traffic of volume of 4500 and 4300 vehicles, respectively. The fraction of heavy-duty vehicles was 8% during the morning and 9% during the afternoon. The amount of traffic heading east was higher during the morning (60%), while, during the afternoon, the amount of traffic heading east was virtually the same as traffic heading west. The mobile laboratory was used to measure particle and $NO_x$ concentrations. Six different distances from roadside along a smaller, nearly perpendicular road were used during both morning and afternoon hours on the same day to gain insight on the concentration changes as a function of distance and time. The measurement distances were 22, 36, 59, 93, 155 and 254 m from roadside and the duration of one measurement was between 4 and 6 min. During the afternoon, the three furthest measurement points had to be repeated resulting in nine data points. Figure 2 shows a map of the measurement location.

To study dispersion, background concentrations of the measured particles and compounds need to be known. Background here means the urban background that is not directly influenced by the road in question. In this study, the background concentrations were measured 680 m downwind from Ring III and one kilometer westwards from the dispersion measurement location with no other major roads nearby.

The mobile laboratory consists of a van equipped with a number of aerosol characterization instruments. The aerosol sample was taken in through a metal tube inlet situated at the front of the mobile laboratory and 2.1 m above the ground. A Particle Size Magnifier (PSM A10, Airmodus, Helsinki, Finland) was used together with Condensation Particle Counter (CPC A20, Airmodus) to measure the total number concentrations of particles larger than 1.3 nm. This combination will be referred to as PSM henceforth. Another CPC (Model 3775, TSI, Shoreview, MN, USA) was used to measure the number concentration of particles larger than 4 nm. The measurement setup and calibration method was similar to that used by Rönkkö et al. [10] during roadside measurements. An Engine Exhaust Particle Sizer (EEPS, TSI) was used to measure particle number concentration, number distribution and mass concentration of particles from 5.6 to 560 nm at 1 s time resolution. The $NO_x$ concentration was measured using a Horiba (Kyoto, Japan) APNA-370 monitor with 60 s time resolution. The mobile laboratory is also equipped with a weather station (WeatherStation 200WX, Airmar Technology, Milford, NH, USA) providing information on wind speed and direction, relative humidity, temperature, air pressure, GPS-coordinates and vehicle speed.

"Sniffer" is Metropolia's mobile laboratory described in detail in e.g., Enroth et al. [16], Pirjola et al. [24] and Pirjola et al. [25]. During our measurements, "Sniffer" was parked 22 m from the roadside and acted as a stationary measurement setup. "Sniffer" was used to measure particles larger than 2.5 nm with a CPC 3776 (TSI). In addition, it was also monitoring the wind conditions using a weather station with a wind sensor (WAS425AH, Vaisala).

Data were collected during the morning from 7:50 a.m. to 8:55 a.m. and during the afternoon from 1:57 p.m. to 3:21 p.m. (Eastern European Summer Time, UTC+3). The temperature was close to 0 °C during the morning measurements and between 3 °C and 5 °C during the afternoon. The wind speed was 1–1.5 m/s during the morning and 1.5–2.2 m/s during the afternoon coming from the direction of the road. Relative humidity during the measurements was 80% during the morning and 68% during the afternoon. The weather during the measurements was overcast with intermittent low-intensity rain, and also some snowfall during the morning.

## 2.2. Data Analysis

The determination of the NCA data is based on the same principle as described in Rönkkö et al. [10], and an identical setup is used in our study. The PSM was run in a constant saturator flow mode, by which the PSM cut-off size is set at 1.3 nm. The CPC used in parallel was a TSI 3776, which has a cut-off size of 4 nm, by which we have determined the NCA size range. A discussion of

the uncertainties of determining the NCA concentration can be found in the Supplementary Materials. In order to account for the differences in response times between the two instruments, 5 s moving averages of the timeseries were used.

The concentrations for each measured or calculated particle size, $NO_x$ and mass was calculated as an average value of the data gathered from each measurement point. The averaging was done over 4 to 6 min time period depending on how long the individual measurement lasted.

A key parameter for dispersion studies is the distance from the aerosol source, in this case the roadside, and the wind speed and direction. To study the dispersion as a function of time, wind data were taken from the stationary setup due to the higher resolution and accuracy of the stationary wind measurements. The average wind speed and direction were calculated from the observations as vector quantities for each measurement point. Using the average wind direction for each measurement, the air parcel travel distance was calculated assuming that an air parcel arrives to the measurement point from the intersection of the wind direction vector and the road. Using this distance and the average wind speed, the travel time of the air parcel was calculated. The average wind speed and direction for each measurement are shown in Figure 2.

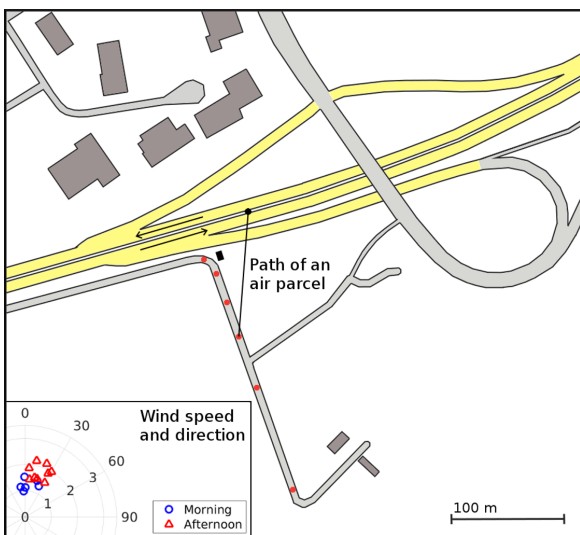

**Figure 2.** The measurements were done downwind from a highway along a small, perpendicular road. The different measurement sites are marked with a red dot. Average wind speed (m/s) and direction during each measurement are shown in the lower left corner. To calculate the time it takes for an air parcel to reach the measurement location, wind angle was used to calculate the point of origin on the highway. The distance between the point of origin and the measurement location was then calculated and wind speed was used to obtain the time between these two locations. The location of "Sniffer" is marked with a black rectangle.

A number of dispersion measurements downwind can be found in literature, studying both the dispersion of aerosol particles and gases such as $NO_x$, CO and $CO_2$. Dispersion studies have been performed e.g., in the USA [18,26,27], in the Netherlands [28,29] and in Finland [16,17,25,30]. These studies show a rapid drop in particle concentrations during the first tens of meters before starting to level to background concentrations beginning from 100–200 m from the measured highway or road depending on the weather conditions and traffic volume. Similar results apply to gaseous pollutants. Enroth et al. [16] found that the dispersion of both particles and gaseous pollutants depends on the surroundings. In an open environment, the concentrations approach the background concentrations more smoothly than in an urban, more built environment. Zhu et al. [18] compared dispersion of aerosol particles, black carbon and CO near two highways with differing traffic fleet compositions. They observed minor differences in dispersion of these pollutants. These were attributed to higher

initial concentrations and different initial particle size distributions next to the highway due to the difference between the ratio of heavy-duty diesel vehicles to gasoline vehicles.

In previous studies, the observed particle concentrations have often been described by finding a function representing the particle concentration by performing a least-square fit to observation. Several fitting methods can be used to the obtained data to describe how dispersion affects the particle concentrations. The options include single term exponential fit $y(x) = a \times \exp(b \times x)$ [16,26,30], single term power series fit $y(x) = a \times x^b$ [17,28] and two term power series fit $y(x) = a \times x^b + c$. We chose to apply a single term exponential fit to model our data because it is often used in existing literature and the parameterization is easy to interpret. It is a simple model to describe dispersion, where parameter $a$ can be considered to describe the initial particle concentration and parameter $b$ describes how rapidly the concentration drops and is therefore dependent on the dispersion and transformation processes affecting the particle population.

## 2.3. Modelling

To simulate the aerosol dynamics occurring during dispersion, we used a quasi-Lagrangian aerosol dynamics model in which the simulated and constantly diluted air parcel moves along a path defined by its travel speed (wind speed) and direction (wind direction) [31]. The model divides the aerosol population in bins of different size ranges to simulate the evolving number distribution of the aerosol population. The model takes into account dilution, deposition, coagulation, condensation and evaporation. Out of these processes, coagulation, condensation and evaporation affect the particle size distribution while dilution, deposition and coagulation reduce the total particle number. Besides reducing the particle number, deposition can also have an effect on the size distribution since it is a size dependent process. The simulation uses the moving sections method, where condensation and evaporation move the bins to different sizes. Coagulation on the other hand can move particles from one bin to another [31].

The equation governing dilution is

$$\frac{dN}{dt} = -D \times N, \tag{1}$$

where $N$ is the particle concentration and $D$ the dilution coefficient.

A similar particle size dependent equation for deposition is

$$\frac{dN}{dt} = -\gamma(d_p) \times N, \tag{2}$$

where $\gamma$ is the deposition coefficient.

To calculate the effect of coagulation, the model uses equation

$$\frac{dN_i}{dt} = \frac{1}{2} \sum_{j=1}^{i-1} K_{j,i-j} N_j N_{i-j} - N_i \sum_{j=1}^{\infty} K_{i,j} N_j, \tag{3}$$

where $N_i$ is the concentration of particles of size $i$ and $K_{i,j}$ is the coagulation coefficient between particles of sizes $i$ and $j$ [32].

To calculate how condensation/evaporation changes the number distribution, the flux of molecules from gas phase to particle phase is calculated and then used to move the sections or particle sizes. To solve the molecular flux from condensing vapour to particle, the continuum regime solution with Fuchs–Sutugin transition regime correction was used [32]. The condensing vapour is assumed to have a very low saturation vapor pressure that is negligible in the condensation flux calculations. The properties of sulphuric acid were used to simulate the condensation of all condensable gases. This gas-to-particle flux was then used to calculate the growth of particles in each size bin.

For modelling purposes, a particle distribution was initialized based on EEPS and NCA measurements. Since EEPS can not measure the smallest particles, the size distribution of NCA was estimated according to measurements done by Rönkkö et al. [10] and combined with the EEPS size distribution. The size distribution goes from 1.3 nm to 200 nm and is divided to 27 size bins. The time step during the simulation was 0.5 s. To study the effect different transformation processes have on different size ranges of the aerosol population, initial values affecting dilution, deposition, coagulation and condensation in the simulation were varied. These values were initial particle concentration and number distribution, dilution and deposition coefficients and the initial condensable gas concentration.

To simulate the particle dispersion, we assumed the highway to be a homogeneous line source with a steady emission rate during the measurements. The mixing and dilution processes are also assumed to proceed uniformly throughout the area during the measurements.

To study the relevance individual processes have on dispersion, one process was neglected while the rest of the processes were still included in the modelling. Dilution and deposition coefficients have the greatest impact on how the concentration falls as a function of time. From a modelling point of view, dilution and deposition work in an identical way since size-dependent deposition was not implemented. If dilution and deposition coefficients are set to zero, particle concentration falls much more slowly and the role of coagulation grows. It is the only process reducing the total particle concentration and has a stronger effect due to larger particle concentration in the absence of dilution or deposition. For an NCA size range, the effect of coagulation is also stronger, but still minor in comparison to that of condensation. The removal of NCA particles by condensation does not seem to be very sensitive to dilution and deposition. While the total particle concentration falls very slowly, NCA concentration falls just as rapidly with or without dilution and deposition. Removing coagulation from the simulation while including the rest of the processes does not have a major effect on total particle concentration or NCA concentration. Both concentrations fall slightly slower, but the difference is minor. When condensation is removed from the simulation, there is no effect on the total particle concentration, but NCA concentration clearly falls slower.

## 3. Results

### 3.1. $NO_x$

The concentration of nitrogen oxides ($NO_x$) is a key parameter for our dispersion study, as it can be considered essentially an inert, dispersing gas without loss processes during the dispersion timescale. While NO and $NO_2$ participate in e.g., ozone formation in the troposphere, their sum $NO_x$ remains constant over the timescales of our study, and its deposition is negligible [32,33]. Due to this, $NO_x$ can be used as a 'clock' for inert, lossless dispersion for which only dilution is a significant process. The $NO_x$ concentration as a function of transport time for both morning and afternoon as well as the exponential fits to data are shown in Figure 3. The background concentrations measured 680 m downwind from the road have been subtracted from the data. One data point 59 m from the roadside was removed as an obvious outlier. The $NO_x$ concentrations are higher during the morning than during the afternoon. This is somewhat unexpected, since the traffic volumes during the morning and the afternoon are similar. The temperatures during the afternoon were slightly higher. This also applies to wind speeds which could cause more efficient mixing and reduce particle and gas concentrations already near the source [26,34]. It has also been seen that in the Helsinki region mixing is stronger during the afternoon [35]. Global radiation data in Helsinki from the Finnish Meteorological Institute [36] show that, while during morning and afternoon measurements the average radiation strength was quite similar, 162 W/m$^2$ and 190 W/m$^2$ respectively, between the measurements, global radiation was much stronger. The average global radiation was 400 W/m$^2$ between 10:00 a.m. and 3:30 p.m. reaching momentarily values up to 900 W/m$^2$. This would reduce atmospheric stability and cause stronger mixing [35].

The roadside concentrations indicated by the fitting curves are 37 ppb during the morning and 23 ppb during the afternoon. These are quite low in comparison to reported values in the Helsinki region. Pirjola et al. [17] reported roadside concentrations up to 170 ppb, but they measured in a more built environment. Enroth et al. [16] reported values in the order of 50–90 ppb in environments more similar to our study. Part of the discrepancy in comparison to the results presented here can be explained by our roadside being slightly further away from the middle of the road than in Enroth et al. [16]. Variation in traffic could also explain some of the differences, but we have no data to confirm this. The afternoon concentrations are not only lower, but they also fall more rapidly, which can be explained by changing weather conditions and stronger atmospheric mixing causing faster dilution.

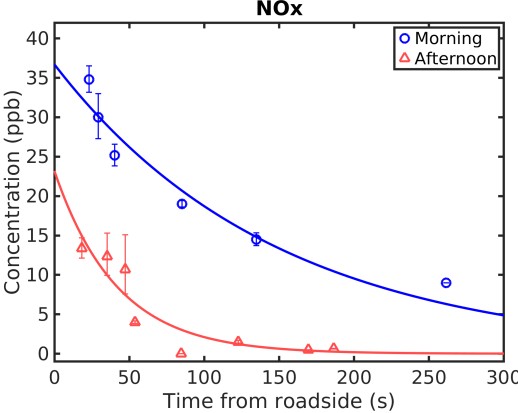

**Figure 3.** $NO_x$ concentrations during morning and afternoon together with exponential fits to data and geometric standard deviations. Background concentrations have been subtracted from the measured values.

Background concentrations of $NO_x$ during morning and afternoon have been presented in Table 1 together with a reference background concentrations from a similar location. The background concentrations from this study were slightly lower than those measured by Enroth et al. [16] but still relatively similar.

**Table 1.** Background concentrations for total particle population ($N_{Tot}$), nanocluster aerosol (NCA) and $NO_x$ compared to reference values from similar measurement location by Enroth et al. [16] and roadside background measurements by Rönkkö et al. [10]. $N_{Tot}$ in this study is the concentration measured with PSM ($d_p > 1.3$ nm) while the reference value was measured with CPC ($d_p > 2.5$ nm).

| Data | | $N_{Tot}$ (#/cm$^3$) | NCA (#/cm$^3$) | $NO_x$ (ppb) |
|---|---|---|---|---|
| This study | Morning | $6.89 \times 10^3$ | $1.98 \times 10^3$ | 8 |
| | Afternoon | $7.29 \times 10^3$ | $2.11 \times 10^3$ | 8.5 ppb |
| References | Enroth et al. [16] | $8.9 \times 10^3$ | | 11–16 |
| | Rönkkö et al. [10] | | $10^3$–$10^4$ | |

*3.2. Total and NCA Number Concentration*

To study the dispersion profile at the roadside, we compared particle number concentrations at different distances and particle size ranges, with urban background concentrations again subtracted from the data. Figure 4 shows the mean concentrations of particles larger than 1.3 nm (PSM), particles larger than 4 nm (CPC) and NCA as a function of transport time from roadside measured during both the morning and afternoon. During the afternoon, the first CPC measurements 254 and 155 m from the roadside failed and had to be repeated, resulting in seven data points for CPC and NCA. The exponential fits (see Section 2.2) to the data are also shown in Figure 4. As can be seen from the figures, the concentrations at the roadside are higher during the afternoon, although the

values from the fits are relatively close to each other and the measured data points actually differ very little between morning and afternoon. Traffic volume and average types of vehicles did not change significantly between morning and afternoon measurements, but it can be speculated that there could have been some momentary changes in the traffic composition affecting the measured concentrations especially closest to the road. The concentration reduction during the afternoon is also faster; this is also visible in the exponential fit and is likely due to faster dilution as with the $NO_x$ results.

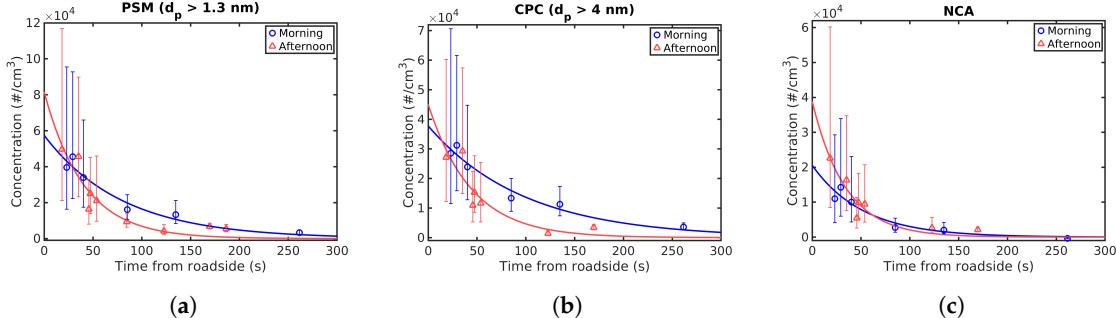

**Figure 4.** The mean concentrations of particles (**a**) larger than 1.3 nm (PSM), (**b**) particles larger than 4 nm (CPC) and (**c**) NCA for morning and afternoon measurements together with exponential fits to data and with geometric standard deviations. Background concentrations have been subtracted from the data. The fitting parameters used are presented in Table 2.

The roadside total particle concentration $N_{PSM}$ obtained from the fits was $5.75 \times 10^4$ #/cm$^3$ for the morning measurement and $8.10 \times 10^4$ #/cm$^3$ during the afternoon. $N_{NCA}$ was $2.06 \times 10^4$ #/cm$^3$ during the morning and $3.87 \times 10^4$ #/cm$^3$ during the afternoon. This corresponds to initial roadside NCA fractions of 36% during the morning and 48% during the afternoon.

The background concentration for particles larger than 1.3 nm was $6.89 \times 10^3$ #/cm$^3$ and $7.29 \times 10^3$ #/cm$^3$ for the morning and afternoon, respectively. For $N_{NCA}$, the respective concentrations were $1.98 \times 10^3$ #/cm$^3$ and $2.11 \times 10^3$ #/cm$^3$. These values together with reference values are presented in Table 1 The fraction of NCA from the total background aerosol population was calculated to be 29% during the morning and 29% during the afternoon. Enough NCA particles therefore survive long enough to contribute a significant part of the background aerosol.

The observed NCA fractions are consistent with earlier reports of urban NCA concentrations. Hietikko et al. [11] measured NCA fractions from 10% to 20% in a street canyon, while, next to a larger road, the NCA fraction was measured to be between 20% and 54% [10]. Rönkkö et al. [10] also observed that the fraction of NCA emitted varied between different vehicles. Measurements in an engine laboratory showed that higher engine loads resulted in higher NCA concentrations and higher NCA fractions from total particle concentration. Similar dependence of sub-3 nm particle concentration on engine loads has also been reported for natural gas engines [37]. Our roadside NCA fractions calculated from the exponential fits in Figure 4 fall roughly between a motorway (up to 50%) and street canyon (10–20%) values, which would be expected for the ring road traffic measured here.

We also compared the CPC measurements from the mobile laboratory with "Sniffer" acting as a stationary setup to ensure that the assumption of a uniform roadside source was reasonable. Figure 5 shows the mobile laboratory CPC concentration as a function of time normalized with the CPC concentrations measured next to the highway with "Sniffer". The CPC aboard "Sniffer" measured particles larger than 2.5 nm, while the mobile CPC measured particles larger than 4 nm. Therefore, the normalization somewhat underestimates the mobile CPC concentration, but the overall trend agrees well with that shown in Figure 4.

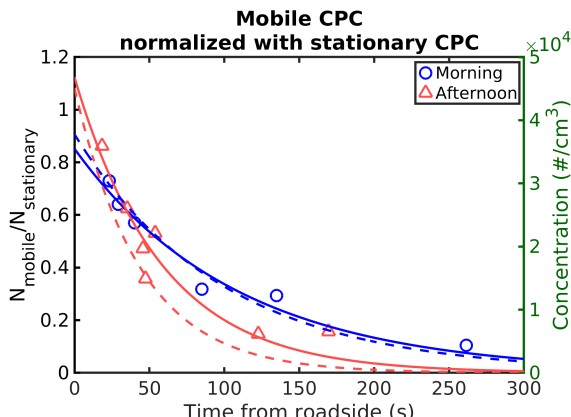

**Figure 5.** The concentration of particles larger than 4 nm measured with CPC aboard the mobile laboratory normalized with the concentrations measured with the CPC aboard "Sniffer" next to the highway. The solid lines are the exponential fits to the normalized data. The dashed lines are the exponential fits to CPC data presented in Figure 4 with their *y*-axis on the right.

### *3.3. Number Distribution*

The particle number size distributions, measured for each distance using EEPS and taking mean values for each measurement points, are shown in Figure 6. The peak concentration for both morning and afternoon measurements closest to the roadside is at 11 nm. In both cases, the concentrations start falling as the time from roadside increases, but the peak does not move. For the afternoon, the under 10 nm particles seem to make a larger contribution as time advances, but this is difficult to confirm using EEPS data due to its size resolution.

Another peak can be seen around 30–40 nm during both the morning and afternoon. During the morning, this peak is more pronounced closest to the road. EEPS data show an emission event while measuring at this location. This event also seemed to be strongest around 40 nm, which could explain why the morning distribution at 23 s differs from other distributions. The number distribution measurements near the roadside are somewhat sensitive to fluctuations in the traffic volume and vehicle types. The concentrations fall as the time and distance from roadside increases. During the afternoon, the corresponding particle size does not change. During the morning, the peak moves to slightly smaller particle sizes, about 25–30 nm.

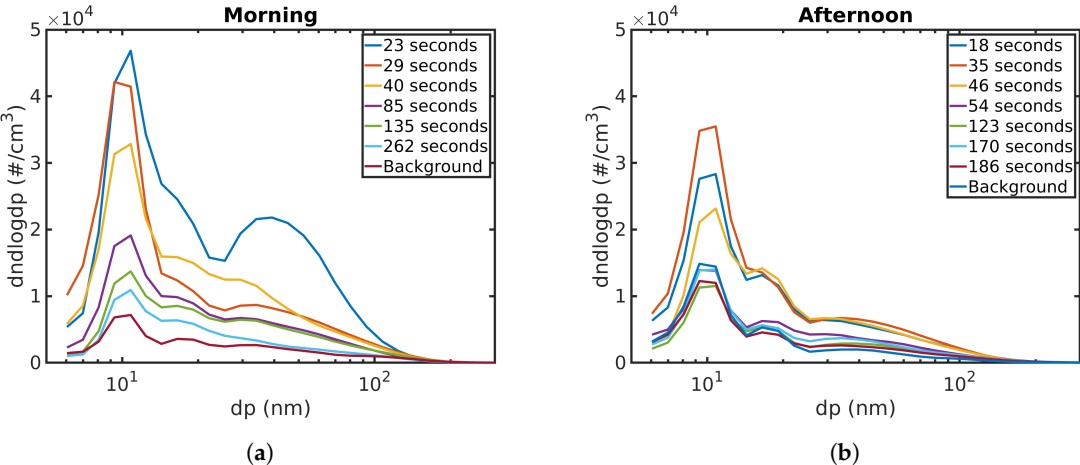

(**a**)　　　　　　　　　　　　　　　　　　　　　　　　(**b**)

**Figure 6.** Number distributions measured with EEPS for different dispersion times during (**a**) morning and (**b**) afternoon.

### 3.4. Mass Concentration

Particle mass concentrations were obtained by measuring volume concentrations with EEPS and assuming a unit density ($1 \text{ g/cm}^3$) and spherical particles. This is based on using aerodynamic diameter as the particle diameter [32]. EEPS is not reliable when measuring particles larger than 200 nm in diameter [38] and therefore only the mass concentration of particles smaller than this was calculated. The mass concentration is therefore quite small. Mass concentrations during the morning and the afternoon as a function of transport time are shown in Figure 7 together with exponential fits to data. Background concentrations were subtracted from the data.

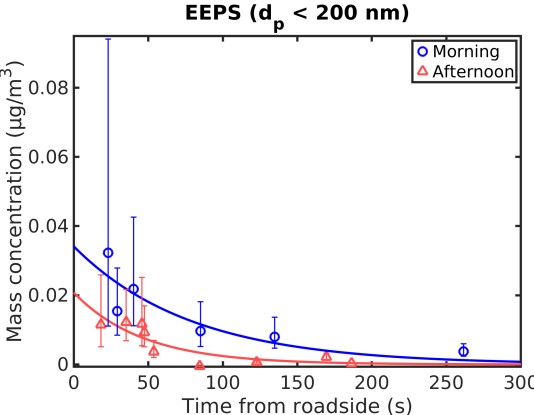

**Figure 7.** Mass concentrations of particles smaller than 200 nm in diameter and exponential fits during morning and afternoon with geometric standard deviations. Background concentrations were subtracted from the data.

### 3.5. Least-Square Fitting of Concentration Profiles

A single term exponential fit $y(x) = a \times \exp(b \times x)$ was used on the data. Parameters $a$ and $b$ are listed in Table 2 for particles larger than 1.3 nm, particles larger than 4 nm and NCA for both the morning and afternoon. Parameter $a$ corresponds to the initial concentration ($\#/\text{cm}^3$) at roadside when the time from the source is 0 s. This suggests NCA fraction of 36% at the roadside during the morning and 48% during the afternoon. The urban background concentrations have been subtracted from the concentration values used here. These values agree well with Rönkkö et al. [10] and support the conclusion that larger roads with higher vehicle speeds emit larger fractions of NCA than what was measured in a street canyon with lower speeds [11]. The measured NCA fraction at 22 m from roadside, or 23 s for morning measurement and 18 s for afternoon measurements, were 31% and 40%, respectively.

Parameter $b$ describes how rapidly the concentration falls as a function of time. A larger negative value indicates the concentration falls faster. For all three size ranges, the concentration drops faster during the afternoon. NCA concentration is also reduced faster than PSM or CPC concentrations. The relative difference between the concentration drop for NCA and total particle concentration (PSM) is larger during the morning measurements. NCA concentration reaches the background value much faster than PSM concentration does during the morning. Because of this, during the dispersion, the NCA fraction goes clearly under the background NCA fraction measured 680 m from the roadside. At some point, the NCA fraction starts increasing and eventually reaches the background NCA fraction as PSM concentration falls to the urban background level. This was not seen in our measurements since, at the most distant measurement location, PSM concentration was still relatively far from the background concentration. For afternoon measurements, this behaviour was not seen since the relative difference between the concentration drop was much smaller (see Supplementary Material, Figure S6).

**Table 2.** Parameters used for single term exponential fits to data for three different size ranges during morning and afternoon. Parameter *a* is the initial particle concentration at the road and parameter *b* describes how rapidly the particle concentration falls.

| Data | PSM ($d_p > 1.3$ nm) | | CPC ($d_p > 4$ nm) | | NCA ($d_p = 1.3$–4 nm) | |
|------|------|------|------|------|------|------|
| Parameter | a | b | a | b | a | b |
| Morning | $5.75 \times 10^4$ | $-0.012$ | $3.78 \times 10^4$ | $-0.010$ | $2.06 \times 10^4$ | $-0.019$ |
| Afternoon | $8.10 \times 10^4$ | $-0.025$ | $4.50 \times 10^4$ | $-0.023$ | $3.87 \times 10^4$ | $-0.029$ |

The normalized mass and number concentrations of particles under 200 nm in diameter measured with EEPS are shown in Figure 8. Both the mass and the number concentrations fall slightly faster during the afternoon. The number concentration here falls slightly faster than the mass concentration.

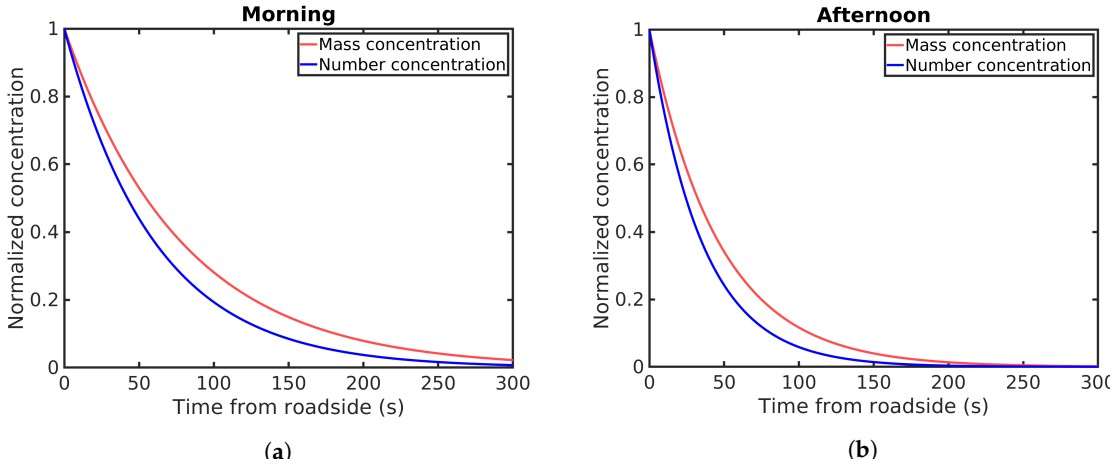

**Figure 8.** Normalized number and mass concentrations measured with EEPS for particles smaller than 200 nm in diameter during the (**a**) morning and (**b**) afternoon. These figures are based on the exponential fits used on the data.

Figure 9 shows the normalized number concentrations for PSM ($d_p > 1.3$ nm), CPC ($d_p > 4$ nm), NCA and NO$_x$. The mass concentration of particles smaller than 200 nm estimated based on EEPS data is also included. The concentrations have been normalized with the initial concentrations at the road obtained from exponential fitting. For particle number concentrations, these initial concentrations are presented in Table 2. The NCA concentration falls to 50% of the roadside concentration after 36 s (50 m) during the morning and 24 s (46 m) during the afternoon. For PSM, these times are 55 s (68 m) during the morning and 29 s (52 m) during the afternoon. Concentrations measured with CPC fall to 50% after 68 s (79 m) during the morning and 32 s (51 m) during the afternoon. This shows that the processes reducing the particle concentration acts faster on the NCA than larger particles for both morning and afternoon conditions. Coagulation has been considered the major process explaining faster removal of small particles [22], as it reduces the amount of small particles while shifting the number distribution to larger particle sizes. However, we measured particle concentrations only for the first few minutes after emission and this is not enough time for coagulation to have a major effect [39,40]. Condensation is another possible mechanism that would have a similar effect: it moves particles to larger sizes, but does not change the total number and also increases the total mass. Evaporation would shift the number distribution to smaller particle sizes while possibly reducing the overall concentration, and reducing the total mass.

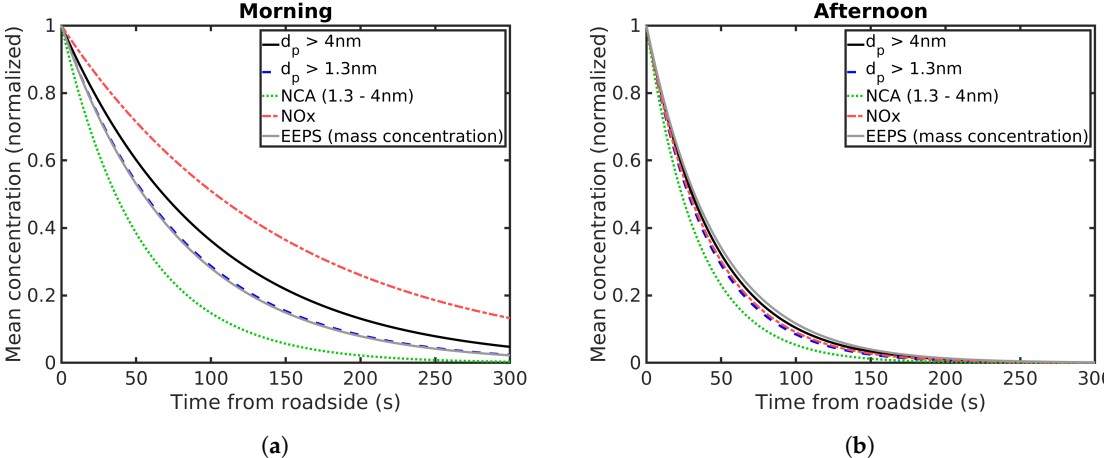

**Figure 9.** Normalized PSM ($d_p$ > 1.3 nm), CPC ($d_p$ > 4 nm), NCA and NO$_x$ concentrations and normalized mass concentrations measured with EEPS during the (**a**) morning and (**b**) afternoon. These figures are based on the exponential fits used on the data.

All of the measured concentrations fall more rapidly during the afternoon than the morning. This is clearly seen in all size ranges, although, with NCA, the difference is smaller than with the other particle sizes and NO$_x$. As was discussed before, on short timescales such as is the case here, NO$_x$ can be considered to be an inert gas. Therefore, the big difference in the NO$_x$ concentration indicates a faster dilution rate during the afternoon probably due to change in weather and global radiation. Higher wind speeds during the afternoon measurements as well as stronger global radiation between the measurements could lead to stronger mixing and thus explain faster dilution. The temperature was also higher during the afternoon.

The behaviour of NO$_x$ in Figure 9 seems to approach that of the particles during the afternoon. This is due to the strong dilution, which decreases the concentration of both particles and NO$_x$ very rapidly. The rates of other transformation processes depend on the particle concentration and therefore strong dilution reduces the relative effect of other possible processes. This explains why the particle and NO$_x$ concentrations are seemingly following the same pattern during the afternoon.

*3.6. Modelling*

For modelling purposes, the dilution ratio due to turbulent mixing was obtained from the NO$_x$ data using Equation (1). The dilution coefficient was $0.64 \times 10^{-2}$ 1/s during the morning and $2.40 \times 10^{-2}$ 1/s during the afternoon. The data for morning was used to study the dispersion of the aerosol. The results are shown in Figure 10. Initially, only the dilution coefficient was included in the simulations and the time development for the total particle number concentration was plotted. It can be seen that dilution alone is not enough to explain the measured data. Condensation has no effect on total particle concentration and turning on coagulation had a very small effect on the simulated particle concentrations. Therefore, another mechanism is needed to explain the difference between measured and simulated concentrations. One such mechanism could be deposition and especially wet scavenging since the weather was rainy. Deposition is a size-dependent process [39,41,42], but, in order the estimate the plausibility of deposition explaining the total particle number data, an average deposition coefficient was used. Including an average deposition coefficient of $0.6 \times 10^{-2}$ 1/s in the simulation fits the measurement data well.

These results would suggest that, in the case of total aerosol concentration, dilution is the main process acting on the total aerosol concentration but other transformation processes than coagulation and condensation with significant impact are also involved. A likely explanation is particle deposition on the ground and possibly wet scavenging by water droplets due to the rainy weather.

To study the dispersion of NCA, dilution and possible deposition were assumed to affect the NCA number concentration the same way it affects the total particle number concentration. In the case

of NCA, dilution and deposition could not explain the measured data. Including coagulation in the simulation had a negligible effect on NCA concentration. The coagulation sink for 1.3 nm particles 23 s from the roadside was $2 \times 10^{-3}$ 1/s (lifetime of ~7 min) and 260 s from roadside $5.3 \times 10^{-4}$ 1/s (lifetime of ~20 min). Including condensation to the simulation, however, affects the concentration reduction significantly. Condensation causes NCA particles to grow to larger sizes, therefore removing particles from the NCA size range while having no effect on the total particle concentration. The data could be explained by using an initial condensable gas concentration of $1.92 \times 10^9$ molecules/cm$^3$. The effect of evaporation is not clear from the modelling studies and, to properly study this, the composition of the particles would have to be known. However, it is possible that some of the NCA particles evaporate completely, thus reducing the NCA concentration.

Implementing size dependent deposition would be likely to have some effect on both total particle concentration drop and NCA concentration drop. Deposition coefficient is larger for smaller particles and therefore the concentration gradient before adding condensation for NCA would most likely be closer to the measured gradient. According to our modelling studies, this seems to have only a minor effect on the necessary condensable gas concentration.

We also looked at the modelled number distributions. Unfortunately, the number distribution measured with EEPS can not measure particles in the NCA size range. Comparing measured and modelled NCA size distributions could give valuable information on the different processes affecting NCA population. The morning peak staying at 11 nm in Figure 6 moves to 15 nm after 300 s in the simulation. This could hint at evaporation playing a role, but it was not possible to confirm this at the time.

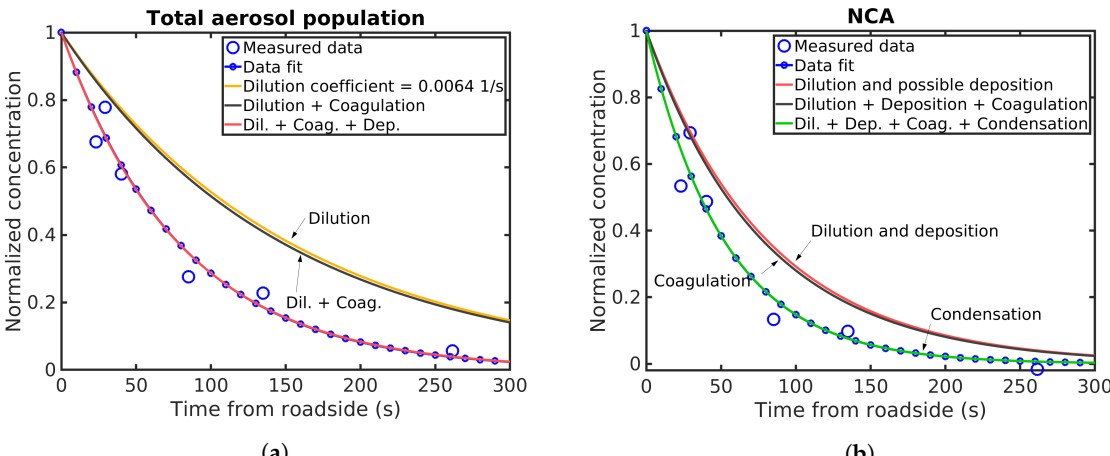

**Figure 10.** Results from the modelling studies for (**a**) particles larger than 1.3 nm (PSM) and (**b**) NCA. For the total particle number concentration, the dilution ratio obtained from NO$_x$ measurements gives concentrations that are clearly above the measured data. Adding coagulation and deposition to the simulation fits the data well for total particle population. For NCA, this is not the case; adding condensation was needed in the simulation to get results that agree with the data.

## 4. Discussion

The overall concentration of particles as a function of distance from roadside depends strongly on dilution. However, modelling studies showed that, to account for the measured data presented here, additional mechanisms are also necessary. Our dataset was limited to one day and specific circumstances, and therefore generalized conclusions should not be made. The morning and afternoon results are also different most likely due to stronger dilution in the afternoon. Of the other transformation processes, condensation has no effect on the number of concentrations of the total particle population. The effect of coagulation was also seen to be very small, which is in agreement with earlier studies [39,40]. The time scales involved are too short for coagulation to have a significant

effect on the aerosol population. Condensation and coagulation therefore cannot explain the measured particle dispersion.

The effect of dry deposition and wet scavenging due to droplets in air is a possible explanation for the measured total particle concentration. This would require an average deposition coefficient of $6 \times 10^{-3}$ 1/s. Dry and wet deposition are strongly affected by the particle size. Dry deposition coefficient can be as high as $5 \times 10^{-3}$ 1/s for particles with 1 nm diameter, falling to $10^{-4}$ 1/s for 10 nm particles and $10^{-5}$ 1/s for 100 nm particles [39,41,42]. The effect of deposition may also be more pronounced near a highway because the high velocities of the vehicles create high friction velocities over the surface of the road [43]. Below-cloud wet deposition coefficient is similarly dependent on the particle size, but also on the intensity of rainfall. Modelling studies suggest that the wet deposition coefficients range from $10^{-4}$ 1/s for 1 nm particles to $10^{-6}$ 1/s to 100 nm particles [44,45]. Deposition coefficient of $6 \times 10^{-3}$ 1/s is therefore high, but plausible considering the large fraction of under 4 nm particles from the total aerosol population and the rainy weather. One should however note that, despite assuming a high deposition coefficient, NCA concentrations still decreased at a higher rate than deposition would suggest. This points towards an additional transformation process playing a major role in the NCA concentration decrease.

Simulating NCA concentration with dilution and deposition gives higher concentrations than the data suggests, as mentioned above. The effect of coagulation is again negligible, but adding condensation to the simulation lowers the NCA concentration significantly. Condensation grows particles out of the NCA range, therefore reducing the NCA concentration, but does not have an effect on the total particle concentration. The morning size distribution in Figure 6 seems to support this, since the concentration peak moves to slightly smaller particle size, which would happen if NCA particles were growing to a larger size range. The simulation was done by defining an initial condensable gas concentration at the highway. The initial condensable gas concentration of $1.92 \times 10^9$ molecules/cm$^3$ fits the data well. This value is obtained by finding a concentration that produces modelling results that fit the data. There are earlier modelling studies suggesting that the condensable gas concentration in the Helsinki region would be between $10^9$ molecules/cm$^3$ and $10^{10}$ molecules/cm$^3$ [46]. The simulated condensable gas concentration value presented here is directly after the emission. Condensation is therefore a likely explanation to the measured data. Size dependent deposition may also explain some of the faster dispersion, since the deposition coefficient used is an average value underestimating the true NCA deposition.

While dilution, deposition, coagulation and condensation can explain the measured data, it is possible that evaporation also has an effect. Evaporation can reduce the NCA concentration if the smallest particles evaporate completely. At the same evaporation could also bring particles from the larger sizes to NCA range. To properly characterize the effect of evaporation would require further studies especially on the composition of NCA and the aerosol population in general. Another mechanism that can have an effect on NCA concentration is nucleation. While in some conditions traffic related NCA production may exceed new particle formation by atmospheric nucleation [10], in other urban environments, it has been seen that atmospheric nucleation has a bigger role than previously though [14]. The relative contribution of traffic-emitted NCA and particles of the same size formed via atmospheric nucleation is not known and most likely changes depending on the urban environment and conditions.

It should be noted that our measurements comprise a limited set of data on NCA dispersion, and they should not be generalized. In our observed conditions, and based on modelling, particle growth is a major factor leading to reduction in concentrations in the NCA size range. The condensing vapours are most likely also originating from the same traffic source as the NCA. In other environmental conditions, this dynamic might change. For example, higher environmental temperatures may shift the saturation concentration of emitted semi- or low-volatility vapours and condensation might not occur. On the other hand, high solar irradiation and associated photochemistry can lead to the formation of condensing vapours both from traffic [47,48] and biogenic [48,49] sources, which could then act as

condensing vapours. In order to determine whether NCA from traffic always grow to larger sizes, further experimental observations at varying distances from the traffic source are needed

The concentration of particles in each size range (1.3–4 nm, larger than 1.3 nm and larger than 4 nm) falls more rapidly during the afternoon in comparison to the morning measurements. Since this also true for $NO_x$ concentrations, it is likely that the change in atmospheric conditions causes faster dilution during the afternoon measurements and reduces the relative importance of particle transformation processes. In our measurements during the afternoon, both temperature and wind speed was higher. Higher wind speed has been seen to reduce particle and gas concentrations already near the source [26,34]. In addition, the global radiation between the measurements was stronger than during the measurements which would reduce the stability of the atmosphere and cause stronger mixing [35].

## 5. Conclusions

We studied the dispersion of nanocluster aerosol or NCA in the size range of 1.3–4 nm as well as the dispersion of the total particle population. Our findings are in agreement with e.g., Rönkkö et al. [10] in showing that the contribution of NCA to the urban background aerosol is significant. A large amount of NCA survives the initial transformation processes after being emitted from traffic. We also found out that NCA dispersion in our observed conditions differed from the total particle population dispersion. Our modelling studies showed that this difference was due to condensation, which reduces NCA concentration by growing particles out of the NCA range but has no effect on the total particle concentration. Size dependent deposition may also explain some of the difference in concentration reduction between NCA and total particle population. We saw that during the first few minutes after the emission, condensational growth to larger sizes was a more significant process in removing NCA than coagulation. This emphasises the role that NCA has in the urban environment, since, rather than being removed by scavenging by larger particles, NCA particles remain in the particle population and grow to larger particle sizes.

Our results apply to specific conditions and should not be generalized, but they do point out that there are differences between NCA and total particle population dispersion. More extensive measurements in different atmospheric and environmental conditions could produce valuable information on NCA dispersion, while possible modelling efforts should include evaporation and take into account the size dependence of deposition. Size distribution measurements in and near the NCA size range could also produce further information on the roles of condensation and evaporation. The effect of nucleation on the NCA population and its relative importance in comparison to traffic-emitted NCA is also unclear and probably depends on the prevailing urban conditions.

**Supplementary Materials:** The following are available online at http://www.mdpi.com/2073-4433/10/6/309/s1, Text 1: Uncertainty in NCA concentration, Figures S1–S5: PSM, CPC, NCA, $NO_x$ and mass concentrations as a function of distance, Figure S6: The fraction of NCA from the total aerosol population, Figure S7: Mass distributions measured with EEPS.

**Author Contributions:** Conceptualization, O.K., H.K., L.P., J.V.N., H.T., T.R. and M.D.M.; Methodology, O.K., H.K., L.P., J.V.N., H.T., S.S., T.R. and M.D.M.; Software, O.K., J.H. and M.D.M.; Validation, O.K., H.K., J.H. and M.D.M.; Formal analysis, O.K.; Investigation, O.K., H.K., S.S., T.R. and M.D.M.; Resources, L.P., J.V.N., H.T. and M.D.M.; Data curation, O.K. and M.D.M.; Writing—original draft preparation, O.K., H.K., T.R. and M.D.M.; Writing—review and editing, O.K., H.K., J.H., L.P., J.V.N., H.T., S.S., T.R. and M.D.M.; Visualization, O.K. and J.H.; Supervision, L.P., J.V.N., H.T., T.R. and M.D.M.; Project administration, L.P., J.V.N., H.T., T.R. and M.D.M.; Funding acquisition, L.P., J.V.N., H.T., T.R. and M.D.M.

**Funding:** The research and results presented are based on work done in an MMEA project, funded by Tekes, and Cityzer project, funded by Tekes (Grant 2883/31/2015), HSY and Pegasor Oy. O.K. would like to acknowledge the Nessling Foundation for financial support.

**Acknowledgments:** The authors are very grateful to Aleksi Malinen from the Metropolia University of Applied Sciences for technical expertise and operation of "Sniffer".

**Conflicts of Interest:** The authors declare no conflict of interest.

## Abbreviations

The following abbreviations are used in this manuscript:

NCA     Nanocluster Aerosol
PSM     Particle Size Magnifier
PSM     Condensation Particle Counter
EEPS    Engine Exhaust Particle Sizer

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
