# Peer review of "Dispersion of a Traffic Related Nanocluster Aerosol Near a Major Road"

_atmosphere, doi:10.3390/atmos10060309_

Round 1
Reviewer 1 Report
General comments
This work investigates ambient concentration levels of nanosized particles near a major road in Helsinki, with particular focus on the so-called nanocluster aerosol (NCA) in the 1.3 - 4 nm size range. Experimental data are collected with state-of-the-art instrumentation and provide an interesting insight on concentration levels in a size range still not deeply investigated. Modelling of particle number concentration downwind the road to match observations is also presented and discussed.
The paper is well structured, clearly written and easy to follow. However, a conclusion session is missing.
The current section 4. Discussion could be renamed as 4. Discussion and conclusion or a fifth section summing up the main points could be added.
Apart from the detailed comments reported below, my greatest concern with this work is the extremely limited extension of the dataset, because measurements were performed for 1 hour in the morning and 1.5 hours in the afternoon. Thus, the overall statistical strength of the presented results is very weak and doesn’t allow to draw general conclusions but only case-specific speculations.
It would be very useful if the authors could enlarge their dataset in order to strengthen their result on NCA concentration levels, to verify the performance of their model for downwind concentration levels, and to consider also other weather conditions (non-rainy days, for instance). Otherwise they should clearly state that their results cannot be generalised without further experimental campaigns.
Additionally, NCA data derive from the difference between concentration levels recorded by different instruments. No mention is made on the uncertainty in these “observed” concentration values, which can be related to instruments accuracy, to measurements principles, cut-off curves shapes. These points should be addressed and discussed more in details.
Finally, there is no mention about number concentration data dispersion around the average values presented, whose averaging time is not specified.
Overall, in spite of the limited dataset, I think that the work deserves publications but the abovementioned points should be discussed.
Keywords. I think that “traffic emissions” is not appropriate for this work. Rather, I suggest “aerosol modelling” or something that give evidence of the effort to match observations with calculated values.
Page 2 line 34: “the first steps of the ultrafine particle number formation process”. I suggest rewording tis sentence into “the first steps of the formation process affecting ultrafine particle number concentration”
Page 2 line 36 “In their study, Rönkkö et al.” For the ease of the reader, add the location of the study
Page 2 line 46 “Upwind concentrations tend to be significantly lower than downwind and in fact differ little from background concentrations” This sentence, but more generally the whole work, needs that the concept of “background” is better explained. Is it urban background?
Additionally, I suggest to add a table with the used background values together with literature values for comparison.
Page 2 line 63 “Delayed primary particles”. Is this a definition given by the authors or does it come from literature?
Page 5 line 171: “is therefore dependant on the transformation processes”. I suggest replacing with “dispersion and transformation processes”
Page 6 line 189: “To calculate how condensation/evaporation changes the number distribution, the flux of
molecules from gas phase to particle phase is calculated and then used to move the sections or
particle sizes”. Please give some more details on the way the gas-to-particle flux is calculated.
Page 6 line 210: As above, the term “background” can be misleading. Here, do the authors intend as background “the concentration “upwind” the road? Otherwise, where was the background measured?
Page 6 line 213-214: It’s true that the ambient temperature in the afternoon is higher than in the morning but for only a few degrees. Maybe, a higher afternoon PBL could be explained by a more intense mechanical turbulence resulting from the higher wind speed. I suggest to revise the sentence also considering this point.
Page 6 line 215-216: “The roadside concentrations indicated by the fitting curves are quite low in comparison to measured values”. It seems that this sentence only relies on the comparison with the reported literature.
Do the authors have road side data for this specific campaign? For the ease of the reader, can they report the literature concentration values they refer to? Finally, do they have an explanation for such mismatch?
Page 7 Fig 3. The authors stated that measurements were done at 6 points with increasing distance from the road. For the afternoon dataset 8 points are represented (6 for morning). Please give explanation about the 2 additional points plotted.
Page 8 line 242-244. Can the authors consider to add a plot reporting NCA fraction vs. road side distance?
Page 8 line 259-260. The size distribution at the distance of 23 m in the morning is clearly different from those of the other monitoring points, with the 40 nm mode much more evident. Can the authors add a comment on this?
Page 8 line 265: “EEPS is not reliable when measuring particles larger than 200 nm”. This sentence here justifies the size range limitation for mass concentration calculations. However, this sentence makes some doubts arise about the reliability of the previous particle number evaluations that refer to a size distribution which goes “from 1.3 nm to 560 nm and is divided to 34 size bins.” (line 195).
Can the authors add comments on this point?
Additionally, can the authors provide justification for their assumption on particle density?
Finally, no mention is made on the mass computation approach. Is it based on spherical particle shape assumption?
Page 8 Figure 7: Here, for the afternoon dataset there are 9 points, differently from Figure 3, and one of these seems to fall below 0. Please explain. I also suggest to rescale the y-axis to positive values only.
Page 9 line 271: I would specify which concentration metric (number, I think) is considered for model fitting.
Page 10 line 287. “measured”: I suggest replacing with “estimated based on EEPS data”
Page 10 line 295-297: “ Coagulation … while shifting the number distribution to larger particle sizes”. However, this change in the number distribution is not evident in Figure 6, where the authors comment that “the peak does not move” in the morning and “corresponding particle size does not change”. Can the authors add comments on this point?
Page 10 line 302-307. I beg your pardon but I don’t understand your explanation for the different behavoiur of NOx in the morning and afternoon sessions. It seems that in the afternoon particles and NOx follow the same decreasing pattern (apparently, atmospheric dispersion acts in the same way), differently from the morning, when NOx decrease less fast than particles. I understand the morning vs. afternoon difference related to weather conditions (wind speed, presumably) but do not find a clear justification for the different decreasing patterns of particles and NOx in the morning. Can the authors add comments on this specific point?
Page 11 line 310: Are these coefficients the b-values for NOx exponential fitting?
Page 11 line 334: “using an initial condensable gases concentration of 1.92 x 109 #/cm3”. What is the metric for this gas concentration? Additonally, what kind of gaseous compounds are included in the family?
Page 11 line 338-339: “The overall … increases”. I suggest to reword this sentence pointing out the different behaviour in the morning and afternoon data. Additionally, I would also clearly state that this conclusion is drawn from just 1 single morning and afternoon campaign, taken in given period of the year and under given weather conditions.
Page 12 line 343: “on the aerosol population”. I suggest to reword the end of the sentence pointing out that here the total particle number concentration is considered, because below condensation is important for NCA, conversely.
Page 12 line 345: “on air”. Maybe, is it “in air”?
Page 12 line 349-350. “The effect … vehicles” I suggest adding an explanation “because of ….”
Page 12 line 356. “concentration decreased” I suggest changing into “concentration still decreased” and in the following “towards an additional transformation process”
Page 12 line 360 “Condensation grows particles out of the NCA range”. Evidence to this sentence can be given by means of Figure 6a where the starting concentration level of the size distribution for 29 s distance is higher than the one of 23 s distance (and 40 s value is similar to 23 s)?
Page 12 line 363 and 365. See comment above on condensible gas concentration metric.
Page 13 line 372: “… measurements” I suggest adding something like “and makes less relevant the impact of particle transformation processes.
Page 13 line 376. While mentioning that evaporation “would need to be taken into account” the authors could add some comments about the potential role of nucleation, which could be a further process to account for under different weather conditions.
Some of the figure captions are too long and replicate the piece of information already given (e.g.: the last sentence in Fig. 4 caption; the caption of Fig. 10)
Author Response
We thank the reviewer for valuable comments that have helped us in improving our manuscript. Our answers to reviewers comments are included in the attached file.

Reviewer 2 Report
Comments are attached

Author Response

(The authors gave the same response as above.)

Reviewer 3 Report
My comments can be found in the attached document.

Author Response

(The authors gave the same response as above.)
